# Tumor-Specific EphA2 Receptor Tyrosine Kinase Inhibits Anti-Tumor Immunity by Recruiting Suppressive Myeloid Populations in Murine Models of Non-Small Cell Lung Cancer

**DOI:** 10.3390/cancers17162693

**Published:** 2025-08-19

**Authors:** Eileen Shiuan, Shan Wang, Dana M. Brantley-Sieders

**Affiliations:** 1Department of Hematology and Oncology, University of California Los Angeles, Los Angeles, CA 90095, USA; 2Medical Scientist Training Program, Vanderbilt University, Nashville, TN 37235, USA; 3Division of Rheumatology and Immunology, Vanderbilt University Medical Center, Nashville, TN 37235, USA; 4Vanderbilt-Ingram Cancer Center, Vanderbilt University Medical Center, Nashville, TN 37235, USA

**Keywords:** EphA2, anti-tumor immunity, non-small cell lung cancer

## Abstract

Given the success of both targeted and immunotherapies against cancer, there is increasing utility for identifying targeted agents that also promote anti-tumor immunity. EphA2 receptor tyrosine kinase has been implicated in the growth of many solid tumors, including lung cancer, and identified as a therapeutic target. Here, we investigate the effects of tumor-specific EphA2 on the anti-tumor immune response in mouse models of non-small cell lung cancer. We found that EphA2 overexpression contributes to increased myeloid cell populations and decreased T-cell populations, as well as decreased T-cell activation, suggesting it creates an immune suppressive environment that allows tumor cells to grow.

## 1. Introduction

Targeted treatments and immunotherapies have both emerged as cornerstones of anti-cancer treatment over the past several decades. Despite these major advancements, many patients do not derive a durable clinical benefit from these therapeutic agents, highlighting the need for more effective strategies [1,2]. One approach is to combine different classes of drugs, such as targeted therapies with immune checkpoint inhibitors (ICIs), to enhance treatment responses. However, to effectively combine these different treatment modalities, more research is needed to evaluate how one may impact the other, in particular, how targeted agents may already be altering the host anti-tumor immune response.

Despite the introduction of many novel therapies, lung cancer remains the most prevalent cancer that occurs among men and women combined and the leading cause of cancer-related death [3]. Only a fraction of patients with advanced non-small cell lung cancer (NSCLC), the most common type of lung cancer, experience clinical benefit from targeted agents or single-agent ICIs, as this is dependent on the mutational profile and programmed death-ligand 1 (PD-L1) expression status, respectively [4]. However, the majority of cases lack actionable alterations and sufficiently high PD-L1 levels; in fact, a PD-L1 tumor proportion score (TPS) greater or equal to 50% is needed for anti-PD-1 or anti-PD-L1 monotherapy, while combination chemotherapy and immunotherapy is the current standard first-line treatment for patients with lower PD-L1 scores [4].

EphA2 receptor tyrosine kinase (RTK) is a cell-surface protein that is overexpressed and implicated in the progression of many solid cancers, including NSCLC, making it a promising target for anti-cancer therapy [5,6,7,8]. For example, translational studies have demonstrated the efficacy of EphA2 kinase inhibitors [9,10] and EphA2-targeted cellular therapies [11,12,13] in preclinical models of NSCLC. A variety of EphA2-targeting strategies, including peptide–drug conjugates, liposomal siRNA, chimeric antigen receptor (CAR)-T cells, and dendritic cell (DC) vaccines, are currently being tested in various cancer clinical trials around the world (NCT05631886, NCT01591356, NCT05631899, NCT06972576, NCT06710158, NCT04180371, NCT05283109, NCT06186401). Although the roles of EphA2 within the cancer cell and tumor endothelium are well studied [14,15,16], their impact on the tumor immune microenvironment is largely unknown [17]. Thus, we aim to evaluate how EphA2 expression in the tumor affects the immune landscape in NSCLC.

To test the impact of tumor-specific EphA2, we overexpressed EphA2 in murine NSCLC cells and found higher tumor burden in syngeneic but not in immunocompromised mice. Flow cytometric analysis of immune infiltrates in tumor-bearing lungs revealed decreased lymphocytic populations and CD8+ T-cell activation, along with increased myeloid populations, including tumor-associated macrophages (TAMs) and monocytes. In addition, transcriptomic profiling demonstrated upregulation of monocyte-attracting chemokines and immunosuppressive proteins. Together, these studies suggest that tumor-intrinsic EphA2 recruits monocytes and promotes their transformation into TAMs, which inhibit the activation of anti-tumor T cells. This provides novel insight into EphA2’s impact on the immune landscape of NSCLC and the additional potential benefits of targeting EphA2 in cancer.

## 2. Materials and Methods

### 2.1. Cell Culture

Lewis lung carcinoma (LLC) cells were a generous gift from Dr. Barbara Fingleton (Vanderbilt University, Nashville, TN, USA). Kras G12D mutant, Tp53 and Stk11/LKB1 knockout, GFP+ NSCLC cell line (KPL) from the C57BL/6 background were previously generated in the lab [8]. Both cell lines were maintained in DMEM (Corning #MT10013CV, Corning, NY, USA) supplemented with penicillin/streptomycin (Gibco #15140163, Grand Island, NY, USA) and 10% FBS (Gibco #A3160502). Luciferase-expressing KPL cells were generated by serial dilutions of KPL cells with lentiviral overexpression of the luciferase gene. Stable EphA2 overexpression was generated by lentiviral transduction of pCDH vector containing the human EphA2 cDNA sequence with subsequent puromycin selection (2 μg/mL) for four days.

### 2.2. Western Blotting

Cells were washed with PBS and lysed on ice with RIPA buffer supplemented with a protease inhibitor cocktail (Sigma-Aldrich #4693124001, St. Louis, MO, USA) and phosphatase inhibitor cocktail (Sigma-Aldrich #4906845001). Lysates were electrophoresed on a 12% SDS-polyacrylamide gel and transferred to nitrocellulose membranes, which were blocked for a half hour with 5% nonfat dry milk in tris-buffered saline with 0.1% Tween-20 (TBS-T) buffer. Membranes were incubated with primary monoclonal antibodies against EphA2 (Cell Signaling Technologies #6997, 1/1000, Danvers, MA, USA) and tubulin (Sigma-Aldrich #T4026, 1/2000) overnight at 4 °C, followed by three washes with TBS-T and incubation with secondary antibodies goat anti-rabbit IRDye 800CW (LI-COR #926-32211, Lincoln, NE, USA) and anti-mouse IRDye 680LT (LI-COR #926-68020, 1/20,000) for one hour at room temperature. After washing with TBS-T, blots were imaged using LI-COR Odyssey.

### 2.3. Cell Viability Assays

For MTT assays, cells were seeded at a density of 1000 cells per well in 100 μL media in 96-well plates. At each indicated time point, 20 μL of 5 mg/mL thiazolyl blue tetrazolium bromide (MTT) reagent (Thermo Fisher #M6494, Waltham, MA, USA) in PBS was added per well and incubated at 37 °C for three hours. Media were then aspirated, and 150 μL of isopropanol solution with 4 mM HCl and 0.1% NP-40 was added to each well and rocked at room temperature for ten minutes. The absorbance at 590 nm was read on a BioTek spectrophotometer and recorded using a Gen5 Microplate Reader. Cell viability was presented as a relative fold change from day-one values. Each assay included six technical replicates and was repeated independently at least three times. For colony formation assays, cells were seeded at a density of 400 cells per well in 2 mL media in 12-well plates. After incubating for seven days, media were aspirated, and plates were washed twice with PBS on ice, fixed with methanol for ten minutes, and stained with 0.5% crystal violet in methanol for ten minutes at room temperature. Plates were then rinsed under diH_2_O and left out to dry overnight before image acquisition. Images were analyzed using NIH ImageJ v1.52o, and colony formation was presented as percentage of total area. Each assay included three technical replicates and was repeated four times.

### 2.4. Animal Models

Animals were housed in a non-barrier animal facility under pathogen-free conditions, 12-h light/dark cycle, and access to standard rodent diet and water ad libitum. Experiments were performed in accordance with AAALAC guidelines and with Vanderbilt University Medical Center Institutional Animal Care and Utilization Committee approval. Wild-type C57BL/6 mice were purchased from Jackson Laboratory and bred to generate litters for experiments. Male athymic nude (Foxn1nu) mice were purchased from Envigo for experiments testing tumor growth and immune infiltrate in the context of T-cell deficiency. All other experiments utilized male and female immunocompetent wild-type C57BL/6 mice. Mice were co-housed with one to four littermates for at least two weeks prior to and during all experiments and compared with littermate controls whenever possible. All mice used for tumor experiments were six to ten weeks old at the onset on the experiment. Experimental cohorts were limited to litters that were born within two consecutive weeks. Sample sizes are as shown in the figures. At experimental endpoints, mice were euthanized by cervical dislocation.

### 2.5. Tumor Models

For lung tumor colonization experiments, luciferase-expressing KPL cells suspended in PBS were injected via tail vein into wild-type C57BL/6 mice, and mice underwent in vivo bioluminescence imaging with a PerkinElmer IVIS Spectrum several hours post-injection to verify successful delivery to the lungs. For experiments comparing lung tumor growth between control and EphA2-overexpressing cells, 1 × 10^6^ KPL cells were injected into each mouse. For experiments equalizing tumor burden, either 4 × 10^6^ vector control or 1 × 10^6^ EphA2-overexpressing KPL cells were injected in each mouse. Mice were reimaged at one and two weeks post-injection and sacrificed on day 14–16. Lungs were weighed, imaged for GFP+ tumors, and processed for flow cytometry analysis. For subcutaneous tumor implantations, 1 × 10^6^ KPL cells suspended in a 1:1 mixture of PBS and Growth Factor-Reduced Matrigel (Corning #354230) were injected subcutaneously into the dorsal flanks of C57BL/6 mice. In experiments with athymic nude mice, tumor dimensions were measured by digital caliper at given time points every other day, and volume was calculated using the following formula: volume = length × width^2^ × 0.52. Tumors were subsequently harvested, weighed, and processed for flow cytometry at day 14 post-implantation.

### 2.6. Flow Cytometry

Lungs and subcutaneous tumors were minced and dissociated in RPMI-1640 media (Corning #MT10040CV) containing 2.5% FBS, 1 mg/mL collagenase IA (Sigma-Aldrich #C9891), and 0.25 mg/mL DNase I (Sigma-Aldrich #DN25) for 45 min at 37 °C. Digested tissue was then filtered through a 70 µm strainer, and red blood cells were lysed using ACK Lysis Buffer (KD Medical #RGF-3015, Columbia, MD, USA). Samples were washed with PBS and stained with Ghost Dye Violet V450 (Tonbo Biosciences #13-863, San Diego, CA, USA) or V510 (Tonbo Biosciences #13-0870) to exclude dead cells. After washing with buffer (0.5% BSA, 2mM EDTA in PBS), samples were blocked in αCD16/32 mouse Fc block (Tonbo Biosciences #70-0161) and stained for extracellular proteins using an antibody master mix made in buffer. After washing with buffer, cells were fixed with 2% PFA. Flow cytometry data were obtained on a BD 4-laser Fortessa using BD FACS Diva software v8.0.1 and analyzed using FlowJo software v10.6.1. Fluorescence minus one (FMO) samples were used as gating controls when needed. Antibodies used in flow panels are detailed in Appendix A, and gating strategies used in analysis are detailed in Appendix A with representative flow plots in Appendix A. Each data point is generated after analyzing at least 5 × 10^5^ viable single cells from a specimen from an individual mouse.

### 2.7. NanoString nCounter Assay

Lung tumors were dissected and minced with surgical tools cleaned with RNaseZAP (Sigma-Aldrich #R2020), and RNA was extracted using RNeasy Micro Kit (Qiagen #74004, Germantown, MD, USA). RNA concentration was assessed with a Nanodrop spectrophotometer, and RNA quality was determined by the Agilent 2100 Bioanalyzer System. Then, 20 ng of RNA from each of twelve mice, six gender-matched littermate pairs, was used for input into nanoString nCounter hybridization and hybridized to the nanoString nCounter Mouse PanCancer Immune Profiling Panel probeset (nanoString Technologies #XT-CSO-MIP1-12, Seattle, WA, USA) to measure gene expression. Raw count data were normalized by background correction, positive control correction, and housekeeping gene correction and log2 transformed using nanoString’s nSolver software v3.0. This software was also used to generate pathway scores, differential gene expression analysis, and the volcano plot. Heatmap was generated with normalized data standardized by gene using Microsoft Excel 2016. Raw data is provided in Appendix A.

### 2.8. RT-PCR

RNA was isolated and measured as detailed above and then converted to cDNA using iScript cDNA Synthesis Kit (Bio-Rad #1708891, Hercules, CA, USA). Quantitative RT-PCR was carried out with TaqMan Gene Expression Assay reagents, specifically TaqMan Fast Advanced Master Mix (Thermo Fisher #4444557) and probes (Appendix A), using a StepOnePlus RT-PCR system (Applied Biosystems, Waltham, MA, USA). Reactions were run in triplicate, and mouse Actb (beta actin) was used as a housekeeping gene. Six gender-matched littermate pairs were used to validate nanoString hits [18], and data were presented as fold change of EphA2-overexpressing tumor samples with respect to their littermate control sample.

### 2.9. Human Gene Expression Correlation Analysis

Publicly available preprocessed RNA sequencing data from The Cancer Genome Atlas (TCGA) lung adenocarcinoma (LUAD) samples were downloaded from cBioPortal [19,20]. Data were log2 transformed before correlation analysis. Data were plotted in Microsoft Excel 2016, and Pearson’s coefficients, R2 values, and *p*-values were calculated. Correlation analyses resulting in R2 greater than 0.1 were considered correlated, and *p*-values of less than 0.001 were considered statistically significant.

### 2.10. Statistical Analysis

All graphs and statistical analyses were completed using GraphPad Prism software v6.07 unless otherwise stated. For comparisons of continuous variables between two groups, an unpaired, two-tailed Student’s *t*-test with Welch correction or unpaired Mann–Whitney U-test was performed. For comparisons of continuous variables over time between two groups, a two-way analysis of variance (ANOVA) was performed. For comparison of survival curves, a log-rank test was performed. For RT-PCR analysis, a one-sample Wilcoxon signed rank test was performed. A *p*-value of less than 0.05 was considered statistically significant unless otherwise stated.

## 3. Results

### 3.1. EphA2 Confers Growth Advantage to NSCLC in Vivo but Not in Vitro 

Given previous work has shown that EphA2 can play both tumor-promoting and -suppressing roles, we investigated whether EphA2 overexpression in our murine NSCLC cell lines impacted cell viability and tumor growth. We chose two NSCLC cell lines originating from the C57BL/6 background, Lewis lung carcinoma (LLC) and a KRAS G12D mutant cell line with concurrent loss of TP53 and STK11/LKB1 (KPL) [8], genetic alterations commonly co-occurring with KRAS mutations in human NSCLC [21]. We stably overexpressed EphA2 in these cells (Figure 1A and Appendix A) but found no changes in cell viability by MTT or colony formation assays in vitro, compared to vector control cells (Figure 1B,C).

We next determined if EphA2 overexpression could impact tumor growth in vivo using two different models: subcutaneous implantation and tail vein injection for generation of lung tumors in immunocompetent, wild-type C57BL/6 mice. In both models, EphA2-overexpressing KPL cells formed significantly larger tumors in vivo, as shown by bioluminescence imaging and gross lung specimens (Figure 1D–F). In addition, survival was significantly worse in mice injected with EphA2-overexpressing cells, compared to control cells (Figure 1G). Overall, these data suggest that EphA2’s pro-tumorigenic effects are mediated by host-specific factors in vivo that are not recapitulated in vitro.

### 3.2. EphA2 Overexpression in NSCLC Does Not Significantly Impact Tumor Burden or Immune Infiltration in Nude Mice

There are several host factors that may explain the discrepancy between our in vitro and in vivo results regarding the impact of EphA2 in NSCLC, one of which is the host immune system. While one prior study demonstrated that tumor-intrinsic EphA2 can inhibit anti-tumor immunity in pancreatic cancer [22], its effect on the tumor immune microenvironment has been largely unexplored. Thus, we set out to determine if the host immune system plays a role in EphA2-mediated KPL tumor growth in vivo.

To test whether the adaptive or innate immune response may play a role, we first evaluated KPL tumor growth in athymic nude mice, which lack functional T cells. In contrast to the pronounced tumor growth observed in immunocompetent mice, we observed no significant differences in tumor growth and weight between control and EphA2-overexpressing tumors in nude mice (Figure 2A). In addition, we performed flow cytometry analysis on the tumors and draining lymph nodes and found no differences in percentages of GFP+ KPL tumor cells and tumor-infiltrating immune cells (Figure 2B). Furthermore, there were no significant differences in the composition of immune cell subsets, including natural killer (NK) cells, B cells, dendritic cells (DCs), and myeloid cells, in either the tumors or draining lymph nodes (Figure 2C,D). These results suggest that EphA2-mediated KPL tumor growth in vivo requires the suppression of T-cell adaptive immunity.

### 3.3. EphA2 Overexpression in NSCLC Decreases Lymphocytic and Increases Myeloid Infiltrate in Tumor-Bearing Lungs

To further investigate the role of the adaptive immune system, we returned to our tumor model with immunocompetent C57BL/6 mice to examine the tumor immune infiltrate using flow cytometry, with a particular focus on T-cell populations. Lungs bearing EphA2-overexpressing tumors had significantly higher percentages of GFP+ KPL tumor cells compared to controls (Figure 3A). Although the overall percentage of immune cells did not differ significantly (Figure 3A), we observed a marked reduction in CD4+ and CD8+ T cells and NK cells in EphA2-overexpressing tumors (Figure 3B). In contrast, myeloid populations, including macrophages and monocytes, were increased, with no notable change in DCs (Figure 3C). These data suggest that tumor-intrinsic EphA2 reshapes the tumor immune microenvironment of the lung, shifting it from a lymphocyte-dominant to a more myeloid-dominant profile. This skewed myeloid response likely contributes to immune suppression and tumor progression by attenuating T-cell-mediated anti-tumor immunity.

### 3.4. EphA2 Overexpression in NSCLC Suppresses Tumor-Infiltrating T Cells

Although we found decreased lymphocyte populations in lungs bearing EphA2-overexpressing tumors, this result may be confounded by the substantially higher tumor burden in these lungs, compared to the controls, which contained less than a percentage of KPL tumor cells. Prior studies have reported a correlation between tumor bulk and the quality of immune infiltrate [23], though discerning which is the cause and which is the effect remains challenging.

To assess if tumor bulk contributes to the immune landscape we observe in Figure 3, we repeated the tail vein experiment while controlling for tumor bulk by injecting four-times the number of control KPL cells relative to EphA2-overexpressing KPL cells. By day 16, both groups exhibited comparable tumor burden (Figure 4A). Interestingly, flow cytometric analysis on these lungs did not recapitulate results from the previous experiment, specifically no decrease in lymphocyte populations in EphA2-overexpressing tumor-bearing lungs (Figure 4B). Notably, CD4+ T cells were increased in the EphA2-overexpressing tumor-bearing lungs. These findings indicate that the differences in immune lung infiltrate we previously observed were at least partially due to increased tumor bulk.

Although the proportion of CD8+ T cells did not differ when tumor burden was equalized between control and EphA2-overexpressing tumors, differences in T-cell activation and exhaustion markers were evident. Tumor-infiltrating T cells with upregulated expression of activation markers, such as CD44, CD69, and CD25, and downregulated expression of exhaustion markers like PD-1 and CTLA-4 indicate a higher T-cell functional status that mediates a stronger and more enduring anti-tumor response [24,25]. EphA2 overexpression in the tumor significantly downregulated surface activation markers CD44, CD69, and CD25 on CD8+ T cells via flow cytometry (Figure 4C). Similar trends were also seen in CD4+ T cells, though these did not reach statistical significance. In addition, tumor-specific EphA2 overexpression led to increased PD-1 expression in both CD4+ and CD8+ T cells, while CTLA-4 levels were unchanged (Figure 4D). These results indicate that EphA2 overexpression in KPL tumors inhibits CD8+ T-cell activation and may promote T-cell exhaustion in the lung tumor microenvironment. Overall, this suggests tumor-specific EphA2 modulates the quality of tumor immune infiltrate, which then subsequently impacts tumor growth.

### 3.5. Gene Expression Profiling Reveals Higher Expression of Myeloid Markers and Chemoattractants in EphA2-Overexpressing Tumors

To elucidate the mechanism by which tumor-specific EphA2 suppresses CD8+ T-cell activation, we performed gene expression profiling of dissected lung tumors using NanoString’s Mouse PanCancer Immune Profiling Panel. Gene pathway analysis yielded two significantly upregulated pathways in the EphA2-overexpressing tumors, cancer progression and macrophage functions, consistent with our prior observations (Figure 5A,B). Differential gene expression analysis revealed 95 genes that were significantly different in expression between control and EphA2-overexpressing tumors with a log2 ratio less than −0.5 or greater than 0.5 (Figure 5C).

Among these differentially expressed genes were monocyte and macrophage surface markers (Csf1r, Ccr2, Itgam/CD11b, Mrc1/CD206), myeloid chemoattractants (Ccl2, Ccl7, Ccl8, Ccl12), and immunosuppressive proteins (Arg1, Tgfb1, Tgfb2, Tgfb3) (Figure 5D). CD11b is a general myeloid marker, while CCR2, CSF1R, and CD206 are typically expressed in inflammatory monocytes and macrophages, with CD206 serving as a canonical marker of tumor-promoting M2-polarized macrophages [26,27]. The CCL chemokines CCL2, CCL7, CCL8, and CCL12 are known ligands for CCR2 expressed on circulating monocytes and facilitate their recruitment into the tumor microenvironment. Additionally, arginase 1 (ARG1) and TGF-β are potent immunosuppressive proteins commonly secreted by tumor-promoting cells such as M2 tumor-associated macrophages (TAMs) and subsequently inhibit CD8+ T-cell effector function. We validated a majority of these genes using RT-PCR (Figure 5E). Correlation analysis using RNA sequencing data from the TCGA further showed that TGFB2 correlated significantly with EPHA2 expression in lung adenocarcinoma samples, though other myeloid markers and immunosuppressive proteins were not significantly correlated with EPHA2 expression (Appendix A).

In summary, gene expression analysis provided evidence that tumor-specific EphA2 increases the levels of myeloid chemoattractants, monocyte/macrophage lineage cells, and immunosuppressive proteins. We propose that EphA2 in the tumor cell upregulates the expression of chemokines in the tumor milieu that recruit circulating monocytes into the tumor, where they differentiate into macrophages and are co-opted to serve tumor-promoting functions as polarized M2 TAMs. These tumor-promoting functions include the secretion of proteins, such as arginase and TGF-β, that inhibit the expansion and activity of T cells, particularly anti-tumor CD8+ cytotoxic lymphocytes. This leads to a shift towards a more myeloid and less lymphocytic infiltrate, as we observed in our flow cytometry studies, as well as decreased activation and increased exhaustion in T cells. Ultimately, the effect of EphA2 in the tumor dampens the anti-tumor immune response and perpetuates the vicious cycle of tumor immune escape and growth.

## 4. Discussion

Our studies identify a novel mechanism that contributes to EphA2’s pro-tumor effect in NSCLC. Although EphA2 has previously been shown to facilitate lung tumor growth and metastasis through tumor cell-intrinsic mechanisms [7,8,9,10], this is one of the first studies that shows an immune-mediated phenotype. Despite these new insights, this work poses several unanswered questions. First, which cell types are critical for the EphA2-mediated effect on CD8+ T cells? EphA2 overexpression in the cancer cell may directly upregulate myeloid-attracting chemokine expression in the tumor cells, but it may also indirectly affect expression in other stromal cells that can secrete these chemoattractants. Although we propose that monocytes and macrophages are responsible for the higher levels of arginase and TGF-β, these immunosuppressive proteins can also be secreted from cancer or stromal cells. Further studies utilizing single-cell techniques and testing specific immune functions will be required to elucidate the roles of each cell type in the tumor microenvironment. Additionally, the molecular mechanisms by which tumor-specific EphA2 alters the chemokine milieu and immune landscape of lung cancer may be very complex. EphA2 in the tumor cell can signal in an ephrin ligand-dependent or -independent manner [28], and it can even be packaged into extracellular vesicles to initiate signaling from a distance [29]. Further molecular investigations are needed to better understand the signaling modalities and downstream pathways in the cancer cell that mediate this immune modulation.

Our findings, overall, align with a study that demonstrated tumor cell-intrinsic EphA2 inhibits anti-tumor immunity in pancreatic cancer through the regulation of PTGS2, the gene encoding cyclooxygenase-2 (COX2) [22]. Markosyan et al. identified EphA2 from a screen of The Cancer Genome Atlas (TCGA) dataset, evaluating genes that were inversely correlated with CD8A expression in human pancreatic adenocarcinomas, as well as other markers of cytotoxic T lymphocytes (CTLs). They went on to evaluate the effect of knocking out tumor cell-specific EphA2 in a mouse model of pancreatic cancer bearing a Kras G12D mutation, loss of Tp53, and a YFP marker (KPCY) and found that EphA2 knockout (KO) significantly increased T-cell infiltration and activation, as we had observed in our murine model of NSCLC. Interestingly, they found that EphA2 KO decreased the numbers of granulocytic myeloid-derived suppressor cells (MDSCs) but did not affect macrophages. Given both KPL and LLC cells are driven by KRAS mutations [30], it is possible that the observed effects on anti-tumor immunity are specific to tumors with a KRAS mutant background. Tumor-intrinsic EphA2 has been shown to have both tumor-promoting and tumor-suppressing [31,32] effects in the presence of KRAS mutations. Whether this dual role uniquely impacts the immune-mediated response in KRAS-driven cancers remains to be seen.

We can draw several parallels between our investigations and the studies performed by Markosyan et al.; however, there are also some intriguing differences. Although both studies demonstrate that tumor cell-specific EphA2 has a detrimental impact on T-cell-mediated immunity, one suggests that granulocytic myeloid cells play an intermediary role, while the other suggests monocytic myeloid cells. Furthermore, Markosyan et al. observed a decrease in tumor cell proliferation and in vivo tumor burden with EphA2 KO, but we observed no differences in growth in vitro or in vivo when we knocked out EphA2 in our KPL model via CRISPR/Cas9 (Appendix A). Lastly, although the authors observed inverse correlations between EPHA2 and CTL gene signatures in human pancreatic cancer, we did not discover any consistent trends between EPHA2 and CD3E, CD8B, GZMA, GZMB, PRF1, or IFNG expression from the TCGA lung adenocarcinoma dataset (Appendix A). These discrepancies could be due to differences in cancer cell type and model and the EphA2 receptor/ephrin ligand balance in the tumor microenvironment, which can be dissimilar between the pancreas and lung. Although EphA2 is implicated in tumor growth and metastasis of many types of solid cancers through ligand-independent signaling [33,34], it can also suppress cell growth via ligand-dependent signaling through ephrin-A1, its primary binding partner [31,35,36]. Because we observed no effect on cell viability in vitro or tumor growth in vivo with EphA2 KO in our KPL lung model, perhaps EphA2 is playing a dual tumor promoter and suppressor role in this cell line, whereas, in KPCY pancreatic tumors, EphA2 is more of a tumor promoter than suppressor. This would suggest that EphA2 is signaling differently in these two tumor types, or at least in these two particular models, and may explain why the immune phenotypes differ.

A note of caution that we feel compelled to point out is that our investigations relied heavily on an overexpression system, which has been shown in published studies to yield artifactual results [37,38,39,40,41] and likely also observed in many unpublished works. We overexpressed other genes in KPL cells, including a catalytically dead Cas9, and found that regardless of the gene, overexpression appeared to increase the tumor burden in vivo, compared to vector control, which appeared to decrease the tumor burden compared to the parental untransduced KPL cells (Appendix A). We then tried to overexpress EphA2 using a retroviral vector and discovered that this unfortunately did not recapitulate results we observed in tumor burden with the lentiviral vector (Appendix A). Discerning which results can be attributed to EphA2 and which to artifact is unfortunately an exceedingly challenging task. 

Despite these conflicting results, we do believe a portion of our work reflects true biology, and not all the data are a result of artifact. In addition to Markosyan et al., another body of work from a different laboratory has recently highlighted EphA2’s role in inhibiting anti-tumor immunity in various solid tumors [42], which provide some evidence of veracity of our studies. Shi et al. reported that EphA2 KO using CRISPR/Cas9 in several syngeneic murine tumor models, including 283LM skin squamous cell carcinoma, 4T1 triple-negative breast cancer, YUMM5.2 melanoma, CT26 colon cancer, and W101 lung adenocarcinoma, led to delayed tumor growth but not in immune-deficient NSG or SCID mice. Using their 283LM model, they show increased T-cell tumor infiltration and activity and reduced MDSCs in EphA2 KO tumors. In addition, single-cell and array analyses demonstrate decreased MDSC-recruiting CXCL1 and CXCL2 and elevated T-cell-recruiting CCL5, CXCL9, and CXCL10 chemokine levels compared to control tumors. Of note, their experiments were conducted in FVB and BalbC mice, while ours were performed in C57BL/6 mice. The difference in cell lines and murine genetic background may partially explain why we did not observe the same phenotype in our EphA2 KO tumors.

The work by our laboratory and others reveals translational insights into the possible immune-mediated effects of current EphA2-targeted therapies undergoing testing in clinical trials, the majority of which of are cellular therapies (NCT05631886, NCT05631899, NCT06972576, NCT06186401) and EphA2-binding drug or toxin conjugates (NCT01591356, NCT06710158, NCT04180371). By targeting EphA2-expressing tumor cells, these treatments likely disrupt downstream signaling and induce cell death, but they may also coincidentally reduce the infiltration of immunosuppressive myeloid populations and increase the infiltration and activity of T cells based on our findings. This would, in theory, increase the functionality of CAR-T cells and may even synergize with immune checkpoint blockade. In fact, two of the ongoing EphA2-targeting cellular therapy trials are testing combinations with immune checkpoint inhibitors (NCT05631886, NCT05631899).

## 5. Conclusions

In summary, our investigations suggest EphA2 overexpression leads to decreased anti-tumor immunity by increasing the recruitment of suppressive myeloid populations and decreasing T-cell infiltration and activation in NSCLC. Our work aligns with the emerging literature that reveals EphA2’s role in immune-mediated mechanisms, in addition to its well-studied tumor-intrinsic mechanisms, that promote cancer progression in multiple tumor types. This not only highlights EphA2’s role in tumor immunology but also provides translational insight into the effects of current EphA2-targeting therapies on the tumor microenvironment. Future studies aimed at further dissecting the specific cell–cell interactions and molecular mechanisms will be critical to fully understand the therapeutic potential of targeting EphA2 and optimize combinations with existing treatment options.

## Figures and Tables

**Figure 1 cancers-17-02693-f001:**
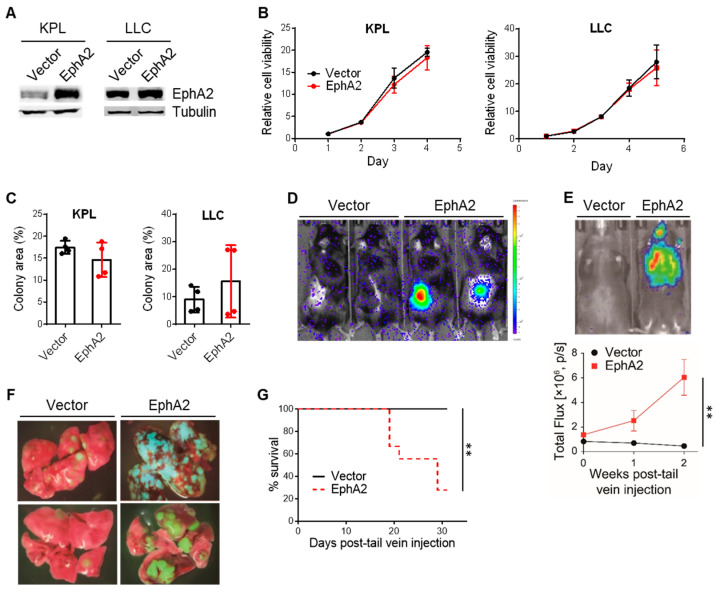
EphA2 confers growth advantage to NSCLC in vivo but not in vitro. (**A**) Confirmation of EphA2 overexpression in KPL and LLC cells by western blot. (**B**,**C**) In vitro cell viability of KPL and LLC cells with control and EphA2 overexpression by MTT and colony formation assays (*n* = 4). (**D**) Representative image of bioluminescence signal in control and EphA2-overexpressing KPL tumors 14 days after subcutaneous implantation. (**E**) Representative image of bioluminescence signal 14 days after tail vein injection of control and EphA2-overexpressing KPL cells and quantification of bioluminescence signal at indicated time points (** *p* < 0.01, two-way ANOVA) (**F**) Representative gross specimens of GFP+ vector and EphA2-overexpressing KPL tumor-bearing lungs. (**G**) Survival of mice injected with vector or EphA2-overexpressiong KPL cells via tail vein (** *p* < 0.01, log-rank test). Data shown are averages ± SD.

**Figure 2 cancers-17-02693-f002:**
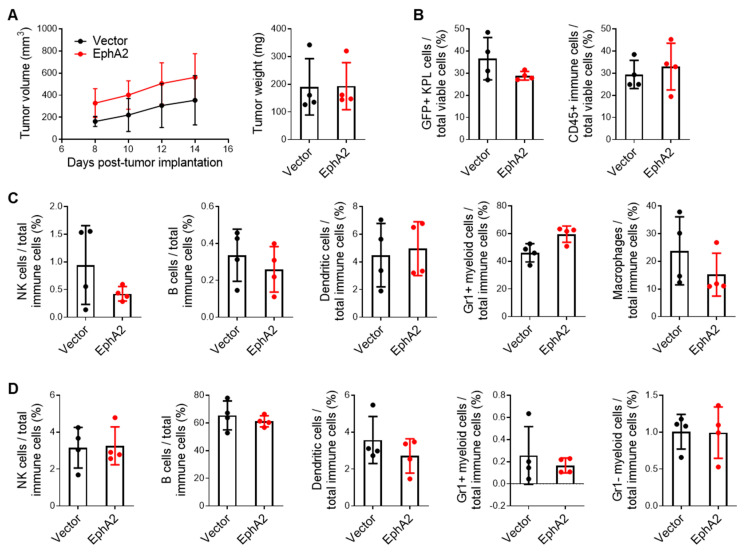
EphA2 overexpression in NSCLC does not significantly impact tumor burden or immune infiltration in nude mice. (**A**) Tumor volumes over time and weights on day 14 post-implantation of control and EphA2-overexpressing KPL subcutaneous tumors from nude mice. (**B**) Flow cytometric analysis of GFP+ KPL tumor cells and total tumor-infiltrating immune cells, as well as (**C**) tumor-infiltrating NK cells, B cells, DCs, macrophages, and Gr1+ myeloid cells. (**D**) Similar flow cytometry analysis of immune populations from draining inguinal lymph nodes. Data shown are averages ± SD (*n* = 4 mice per group).

**Figure 3 cancers-17-02693-f003:**
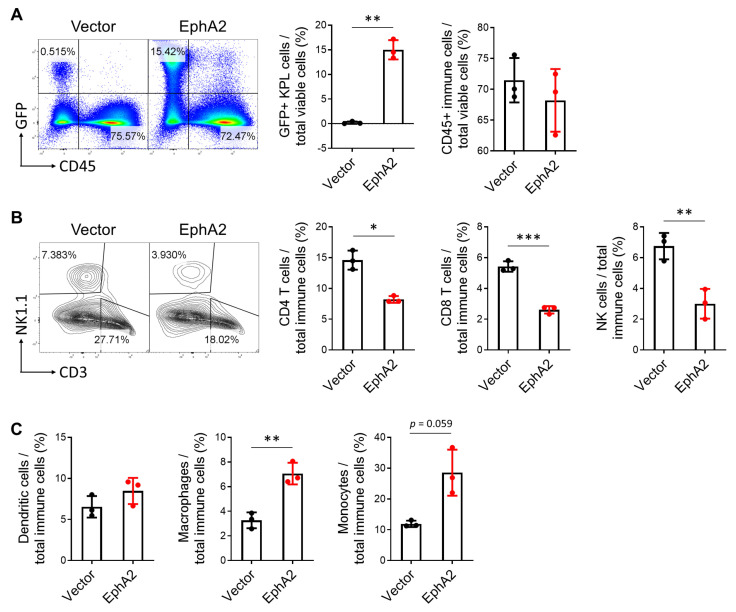
EphA2 overexpression in NSCLC decreases lymphocytic and increases myeloid infiltrate in tumor-bearing lungs. (**A**) Representative flow cytometry plots and quantification of GFP+ KPL and immune cells from vector control and EphA2-overexpressing tumor-bearing lungs on day 14 post-tail vein injection. (**B**) Similar flow plots and analysis of CD4+ and CD8+ T cells and NK cells, as well as (**C**) quantification of DCs, macrophages, and monocytes. Data shown are averages ± SD (*n* = 3 mice per group, * *p* < 0.05; ** *p* < 0.01; *** *p* < 0.001, two-tailed unpaired Student’s *t* test with Welch correction).

**Figure 4 cancers-17-02693-f004:**
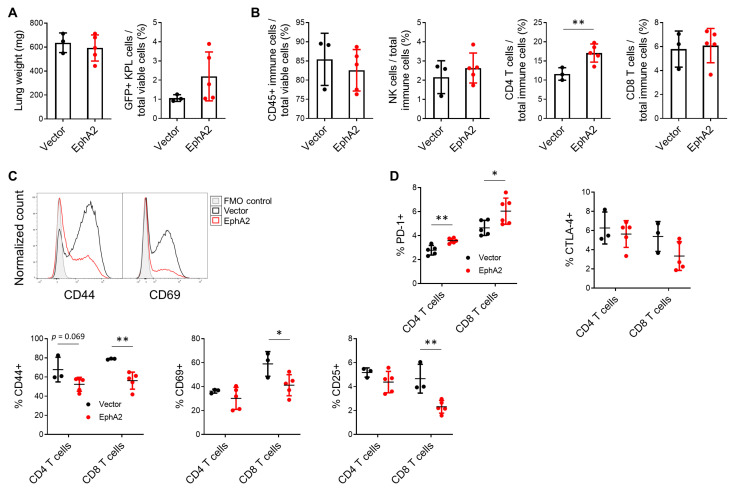
EphA2 overexpression in NSCLC suppresses tumor-infiltrating T cells. (**A**) Lung weights and quantification of GFP+ KPL cells via flow cytometry from vector control and EphA2-overexpressing tumor-bearing lungs with equalized tumor burden. (**B**) Flow cytometric analysis of total immune cells, CD4+ and CD8+ T cells, and NK cells in KPL tumor-bearing lungs. (**C**) Representative flow histograms of CD44 and CD69 expression on CD8 T cells and quantification of CD44, CD69, and CD25 activation markers on CD4 and CD8 T cells. (**D**) Quantification of PD-1 and CTLA-4 exhaustion markers on CD4+ and CD8+ T cells. Data shown are averages ± SD (*n* = 3–6 mice per group, * *p* < 0.05; ** *p* < 0.01, two-tailed unpaired Student’s *t* test with Welch correction).

**Figure 5 cancers-17-02693-f005:**
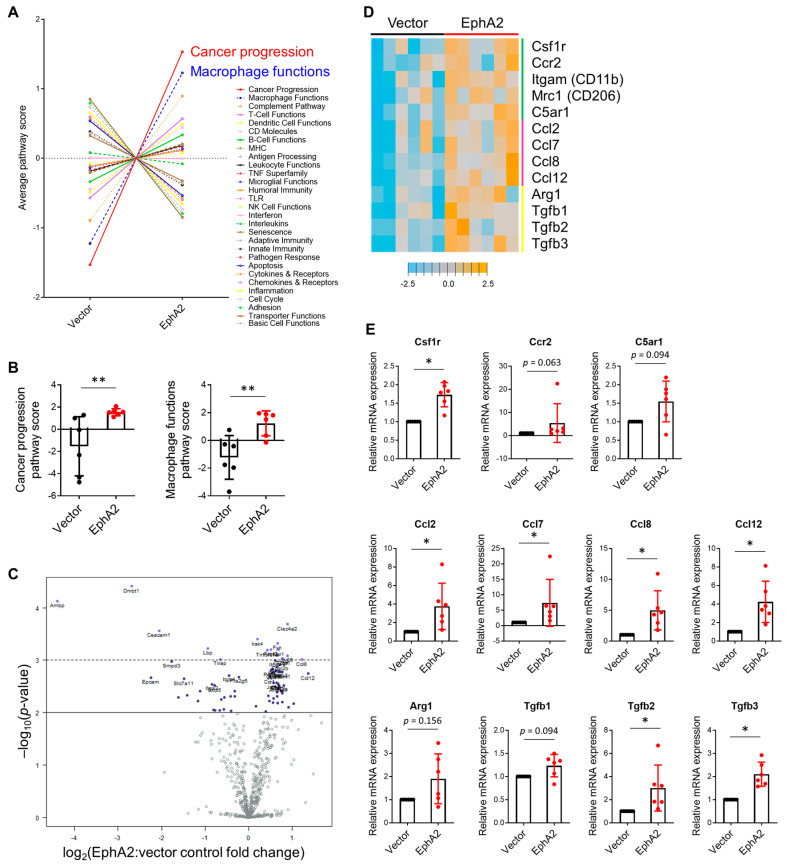
Gene expression profiling reveals higher expression of myeloid markers and chemoattractants in EphA2-overexpressing tumors. (**A**) Average pathway scores of vector control and EphA2-overexpressing KPL tumors calculated from normalized gene expression data using nanoString nSolver software. (**B**) Comparison of cancer progression and macrophage functions pathway scores between control and EphA2-overexpressing samples. (*n* = 6 mice per group, ** *p* < 0.01, unpaired Mann-Whitney test) (**C**) Volcano plot of statistically significant differentially expressed genes. (**D**) Heatmap depicting standardized expression of differentially expressed myeloid markers (green bar), myeloid-attracting chemokines (pink bar), and immunosuppressive proteins (yellow bar). (**E**) RT-PCR validation of nanoString hits. (*n* = 6 mice per group, * *p* < 0.05, one-sample Wilcoxon signed rank test). Data shown are averages ± SD.

## Data Availability

All raw data is publicly available at Harvard Dataverse: https://dataverse.harvard.edu/dataverse/EphA2tumorimmunityinNSCLC (accessed on 18 August 2025).

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
