# Peer review of "Tumor-Specific EphA2 Receptor Tyrosine Kinase Inhibits Anti-Tumor Immunity by Recruiting Suppressive Myeloid Populations in Murine Models of Non-Small Cell Lung Cancer"

_cancers, 2025, doi:10.3390/cancers17162693_

Round 1
Reviewer 1 Report
Comments and Suggestions for Authors
The article Tumor-Specific EphA2 Receptor Tyrosine Kinase Inhibits Anti-Tumor Immunity by Recruiting Suppressive Myeloid Populations in Murine Models of Non-Small Cell Lung Cancer by Shiuan et al. presents compelling evidence that tumor-specific overexpression of EphA2 modulates the tumor immune microenvironment in NSCLC by promoting a suppressive myeloid milieu and inhibiting T cell-mediated anti-tumor immunity. The study is well-structured and methodologically thorough, employing a combination of in vitro assays, murine syngeneic models, flow cytometry, NanoString profiling, and RT-PCR validation. It addresses a significant gap in the understanding of how EphA2 influences tumor–immune interactions, extending prior observations in other cancer types to the context of lung cancer. Additionally, the role of EphA2 in immune suppression is underexplored in lung cancer. The manuscript provides a novel mechanistic insight into how EphA2 may contribute to immune evasion, a key hallmark of cancer.
Minor /Specific Comments and Suggestions:
Introduction: The background on EphA2 is well-framed, but the rationale for focusing on lung cancer models could be further emphasized by citing epidemiological or translational studies.
Results:
The study would benefit from integrating human data (e.g., analysis of TCGA or other transcriptomic datasets) to assess whether EPHA2 expression correlates with myeloid infiltration, immune suppression markers, or clinical outcomes in NSCLC. The authors mention a lack of consistent correlation in TCGA but provide no data, even in supplementary material.
Including representative flow cytometry gating strategies in the main figures would enhance reproducibility.
Please clarify/discuss whether the increased tumor burden in EphA2+ tumors contributes causally to immune changes or is a consequence of them (especially in Figure 3 vs. Figure 4).
Discussion:
The authors could expand on how EphA2-targeted therapies (currently in clinical trials) might be affected by these findings—e.g., whether combining with immune checkpoint blockade is a viable strategy.
As a suggestion, also consider whether the observed effects differ by tumor subtype or molecular background (e.g., KRAS, STK11 mutations).
Supplementary Data: It would be valuable to include RNA-seq/NanoString full gene expression tables and gating strategy figures in supplementary files.
Reviewer 2 Report
Comments and Suggestions for Authors
In this paper, the authors investigate effect of RTK EphA2 in anti-tumor response of lung cancer (NSCLC) in mouse model. To test the impact of tumor-specific EphA2, they overexpressed EphA2 in murine NSCLC cells.
Intro 1st para should have relevant lit cited
line 87 - origin of KPL cells? there is another breast cancer cell line of this name...so better use some other terminology for clarity
line 88 - phosphatase inhibitors should be singular (as it's only 1 chemical), TBS-T composition?
line 184- mouse Actb?? need full form...fig1 A shows tubulin as housekeeping??
line 188 - needs a cited suitable ref
Fig 1 F- what is diff between top and bottom panel?
Conclusion should be more detailed.
ethical approval for animal studies should be mentioned.
Reviewer 3 Report
Comments and Suggestions for Authors
Page 2, line 45: i would change "includes designing the best combinations of targeted therapies and immune checkpoint inhibitors (ICIs)" with "One possible approach is to combine different class of drugs such as immune checkipoint inhibitors with targeted terapies to enhance immune-response";
Page 2, line 51: insert a citation for the afffirmation that lung cancer remain the most prevalent cancer;
Page 2, line 57: I would change that the majority of cases lack sufficient high PD-L1 levels: in fact, the PD-L1 expression is needed for anti pd-l1 in monotherapy, combination therapy is the standard ros low pd-L1 expression;
Page 12, line 42: personal consideration of the authors, i wuold delete it;
Page 12, line 44: I would delete the last affirmation ("But also offers a cautionary tale of scientific rigor and reproducibility".
